# Identifying Critical Indicators in Performance Evaluation of Green Supply Chains Using Hybrid Multiple-Criteria Decision-Making

Changlu Zhang [1,2], Liqian Tang [1,2] and Jian Zhang [1,2,*]

1 School of Economic & Management, Beijing Information Science & Technology University, Beijing 100192, China; 20151935@bistu.edu.cn (C.Z.)

2 Beijing Key Lab of Big Data Decision Making for Green Development, Beijing Information Science & Technology University, Beijing 100192, China

* Correspondence: zhangjian@bistu.edu.cn

**Abstract:** Performance evaluation of green supply chains (GSC) is an important tool to improve their comprehensive management. Identifying critical indicators is crucial to evaluation. This study examines the critical indicators in performance evaluations of GPC and provides relevant suggestions for managers to improve GSCs' performances. Firstly, we summarized 24 evaluation indicators from five dimensions—financial value, customer service-level, business processes, innovation and development, and the so-called green level. Secondly, the Delphi method was used to determine the formal research framework. The fuzzy decision-making trial and evaluation laboratory based analytic network process (fuzzy DEMATEL-based ANP) model was applied. The weighted prominence of each indicator was calculated to identify those that were critical, and a causality diagram was constructed for them. Finally, corresponding countermeasures and implications regarding those were put forward. The research results show that the critical indicators include the return rate of net assets, the growth rate of profit, the rate of service satisfaction, market share, production flexibility, and the green consensus. Among them, the green consensus, the growth rate of profit and the rate of service satisfaction form a virtuous circle, leading to the improvement of the overall performance of GSC.

**Keywords:** green supply chain; performance evaluation; critical indicators; fuzzy DEMATEL-based ANP; MCDM



## 1. Introduction

A supply chain is a customer-oriented organizational form that aims to improve quality and efficiency by integrating resources for efficient collaboration in product design, procurement, production, sales, and service. In recent years, with the increasingly severe global climate problem, the construction of a green supply chain (GSC) has received increased attention. Based on the traditional supply chain, the GSC integrates concepts of sustainable development such as green manufacturing, product life-cycle management, and producer responsibility for enterprise value-chain activities. The GSC comprehensively considers the economic, social, and ecological benefits of enterprises through the entire supply chain process from raw material acquisition, processing, packaging, storage, transportation, use and recycling, and green sustainable development [1,2]. The GSC has the characteristics of integrity, purpose, hierarchy, environmental adaptability, and complexity [3].

Performance evaluation of the GSC is important for its construction and management. Since the GSC is based on the concept of the whole life cycle, its performance evaluation pertains to all aspects of planning, procurement, production, logistics, marketing, and recycling [1]. The GSC management process is more complex than that in traditional supply chains because the GSC involves longer evaluation cycles with diverse evaluation attributes. As such, evaluating its performance is challenging.

The traditional methods used for evaluating supply chain performance include the supply chain operation reference (SCOR) model and the balanced scorecard (BSC) model. The SCOR model is an effective tool for diagnosing the performance of a GSC [4]. The model is vertically divided into four levels (the process level, configuration level, practice level, and implementation level) and horizontally divided into five links (planning, procurement, production, distribution, and return management) [5]. The SCOR model is usually process-oriented, and the efficiency of the process is considered the key to the efficiency of the enterprise, which conforms to the GSC business framework. However, the model does not analyze the overall supply chain participants, and it does not take into account the indicators of all links of the supply chain [6]. The BSC model can explain the relationship between strategy and process through four dimensions: finance, customers, internal processes, learning, and growth [7]. The model comprehensively measures the balance between financial and non-financial, long-term and short-term, cause and result, internal and external, and qualitative and quantitative indicators [8]. It provides a comprehensive framework for measuring performance. However, the BSC model is guided by the overall strategy and focuses on the comprehensive development of enterprises. There are defects in process-oriented organizational performance evaluations of supply chains [9].

Thus, whereas the BSC model provides a basic framework for GSC performance evaluations from a macro-perspective, the SCOR model provides clear business divisions for the GSC. The combination of the two models can thus make the construction of an evaluation index system more comprehensive and scientific [10]. To construct such a GSC performance evaluation index system, we introduce a combined BSC-SCOR model that merges the evaluation dimension with the operation process. On the one hand, according to the core idea of the BSC model, we introduce a multi-dimensional index, and on the other hand, business processes are divided and decomposed according to the SCOR model [11].

Having constructed the evaluation index system, we identify the key indicators of performance evaluation of a GSC. The interaction between these indicators plays a vital role in improving performance. The identification of key indicators of a GSC performance evaluation is a typical multi-criteria decision-making (MCDM) problem. For this kind of problem, commonly used methods include the analytic hierarchy process (AHP) method, the entropy weight method, the analytic network process (ANP) method, and the decision-making trial and evaluation laboratory (DEMATEL), among others. However, the AHP and entropy weight method are only applicable when the indicators are independent of each other, without considering the interaction between them. In reality, there are interactions between different types of performance evaluation indicators. In the ANP model, each indicator has a network structure that affects other ones, and this method offers a consistency check and complex modeling for comparing indicators [12,13]. The DEMATEL method calculates the causality and prominence of each indicator to determine the key indicators. It then builds a causal relationship diagram of the indicators to reflect the interaction of each indicator. The total influence matrix and causality diagram obtained by the DEMATEL provide basic data for ANP, thus avoiding the consistency test and complex modeling problems that arise from only using the ANP. In our study, to capture the ambiguity and uncertainty of various indicators, we combine the fuzzy DEMATEL method with the ANP method to form the fuzzy DEMATEL-based ANP model to identify the key performance evaluation indicators for a GSC [14,15].

In our study, we explore the answers to the following research questions: What dimensions and indicators should be considered in the performance evaluation of GSC? Which are the critical indicators and how do they affect each other? Furthermore, which are the critical driving factors that affect the GSC's performance most? Answering these questions has certain theoretical and practical implications. Theoretically, we sort out the performance evaluation indicators of a GSC systematically and comprehensively by combining the BSC and SCOR models. Furthermore, we analyze the interaction between any two indicators and identify the critical driving indicators with the adoption of the

fuzzy DEMATEL-based ANP model. Practically, as shown in the case study, the findings are conducive to scientific evaluation and effective improvement of a GSC's performance.

The rest of this paper is arranged as follows: Section 2 reviews and summarizes previous research results on GSC performance evaluation. In Section 3, we introduce the Delphi method and the fuzzy DEMATEL-based ANP model used in the study. Section 4 presents our empirical research, which selects and analyzes the key performance evaluation indicators of a GSC. Section 5 discusses relevant countermeasures and offers suggestions for performance evaluation of GSC based on the empirical research results. Section 6 concludes.

## 2. Literature Review

A GSC is essentially an endorsement by green suppliers that the green attributes of supply chain management processes have been satisfied. They are an important means of optimizing the market environment, establishing a trust mechanism, and developing international green trade. Evaluations of GSC performance are a powerful guarantee for ensuring the greenness and effectiveness of the whole supply chain operation process. The GSC is a hot topic in the field of supply chain research. As the world continues to advocate for environmental awareness, the issue of evaluating GSC performance is also being studied and expanded upon by scholars.

On the selection of GSC performance evaluation indicators, scholars have conducted in-depth research from different perspectives. One focus is the supply chain business process, which includes all aspects of the supply chain operation process. Business process performance evaluation indicators are extracted according to the construction principles of the supply chain process performance evaluation indicator system. Liao analyzed the links at all levels of the resource input level, operation level, product (service) output level, and feedback level, and then screened out specific evaluation indicators [16]. Feng and Li constructed corresponding indicators from three aspects: the result level, operation level, and support level. They used the entropy method and AHP method to determine the weight of indicators for evaluating performance [17]. Jin selected 21 indicators from four dimensions related to the supply chain process to conduct research on evaluating the GSC performance of automobile enterprises in the carbon peak and carbon neutral backgrounds [18]. Osintsev et al. conducted a detailed study on the performance evaluation of GSC logistics processes based on a combined DEMATEL-ANP method to achieve the goal of sustainable development [19]. Effendi et al. and Divsalar et al. used the SCOR model to evaluate the performance of GSCs [20,21].

Another focus of research is capital operations and the financial management requirements of supply chain enterprises. Hou and Wang respectively analyzed the financial performance of A-share listed logistics supply chains and forestry logistics enterprises and built an enterprise financial performance evaluation index system based on four dimensions: enterprise profitability, debt paying ability, operating ability, and development ability [22,23]. Based on the concept of green development, Liu et al. added indicators that reflected low-carbon capacity to the four traditional dimensions of enterprise financial evaluation systems. They focused on measuring the efficiency of energy consumption, pollutant emissions, and post-production waste recycling [24]. Based on the concept of green development, Yu and Li built a comprehensive financial analysis index system that includes five dimensions: solvency, operating ability, profitability, development ability, and environmental protection and governance ability [25].

From the perspective of ecological civilization, some scholars argue that the impact of the whole life cycle of products on the environment should be considered in addition to economic and social benefits. Chang et al. started from the external value chain of enterprises, and comprehensively analyzed the impact of the upstream suppliers and downstream vendors (customers) outside the enterprise. Based on this, they built a set of performance evaluation indicators to comprehensively manage the environmental performance of enterprises [26]. Wang et al. extracted 20 environmental performance evaluation indicators from six perspectives: ecological design, cleaner production, resource and energy utilization,

waste recycling, environmental impact, and financial performance related to environmental activities [27]. Yu et al. analyzed the environmental performance of automobile enterprises from both internal and external aspects [28]. Zhou et al. evaluated green indicators such as green design, green procurement, and the green production of enterprises based on the Delphi and fuzzy AHP methods [29]. Huang et al. used AHP and three-stage DEA methods to conduct comprehensive performance evaluation research on China's energy supply chain in the context of double carbon goals [30]. Wicher et al. used multi-criteria decision-making to evaluate the sustainability performance of industrial enterprises [31].

There has also been extensive research done on various supply chain performance evaluation indicators, and on refining performance evaluation indicator systems. Some of this research pertains to the selection of critical performance evaluation indicators. In research on performance evaluation indicator segmentation, Wan, Xiao, and Gu considered the perspective of corporate stakeholders and added sustainable development indicators on the basis of the four dimensions: the financial situation, customer service, operation process, and innovative learning. Their green development indicators were subdivided into five categories: environmental governance investment, the environmental protection investment ratio, the environmental protection material utilization rate, the resource utilization rate, and the resource recovery rate. Based on these indicators, they established a GSC performance evaluation index system [32–34]. Du studied the performance evaluation of GSC from four dimensions—finance, customers, processes, and sustainable development—and subdivided each aspect (e.g., dividing sustainable development into new product development capability, technology investment, and the community environmental protection level) [35]. Wang and Lu, Bai et al. and Huang et al. considered the target benefits of GSC combined with the actual operations of GSC. They built a performance evaluation index system based on three aspects—economic benefits, social benefits, and environmental benefits—and on this basis, subdivided these into a secondary evaluation index system for comprehensive evaluations [36–38]. Wang and Yang studied the characteristics of green agricultural product transportation and built a green agricultural product supply chain performance evaluation index system with six dimensions: financial status, customer service, business processes, the logistics technology level, innovative learning, and green environmental protection. In order to meet the requirements for agricultural product transportation, the logistics technology level was subdivided into four aspects: delivery timeliness, loss rate, service flexibility, and order completion rate [39].

In terms of selecting critical performance evaluation indicators, Zhang et al. used the fuzzy technique for order preference by similarity to an ideal solution (TOPSIS) method to evaluate the performance of the GSC [40]. Jiao used the multi-objective evaluation analysis method to construct an indicator system and used the fuzzy comprehensive evaluation method to evaluate the environmental management performance of GSC [41]. Cao and Fan studied the performance evaluation of green agricultural product supply chains based on DEA and principal component analysis [42]. Ma et al., constructed a GSC performance evaluation index system from the aspects of process, customer service, finance, environmental protection, information, and knowledge, and proposed a method to identify the critical performance evaluation indicators of GSC based on the DEMATEL method [43]. Dai and Ye adopted a low-carbon perspective and analyzed and evaluated the key indicators of GSC optimization and supervision from five dimensions regarding property value and internal supply chain processes [44]. Nozari et al., conducted quantitative analysis on critical performance evaluation indicators of GSC in the fast-moving consumer goods industry based on a nonlinear fuzzy method [45]. Chang et al., used the mixed MCDM model to analyze the supply chain performance indicators of Indian mining and earth-moving equipment manufacturing companies through the DEMATEL method [46].

The above research results provided certain guidance for the construction of the initial evaluation index in our study. According to the principles of systematicness, comprehensiveness, a combination of static and dynamic indicators, and a combination of qualitative and quantitative indicators, we integrated and optimized the indicators involved in pre-

vious studies considering the actual requirements of a performance evaluation of a GSC. Firstly, referring to the BSC model, we summarized the previous evaluation indicators into four dimensions: financial value, customer service level, business process, innovation and development. Secondly, considering the characteristics performance evaluation of the GSC, indicators related to green, low-carbon, and sustainable aspects were summarized into the dimension of green level. Thirdly, the supply chain is the integration of all processes and activities from the initial supplier to the final customer. In order to reflect the characteristics, the business process was further decomposed in combination with SCOR model. Finally, the initial performance evaluation index system of the GSC was established as shown in Table 1.

**Table 1.** The initial performance evaluation index system of the GSC.

| Dimension | Indicator | Indicator Description | References |
|---|---|---|---|
| Financial value | Return rate of net assets | Enterprise profitability | [22–24,26,34,35] |
| | Return rate of total assets | Overall profitability of all assets including net assets and liabilities | [17,24,33,34,37,39] |
| | Total asset–liability ratio | Ability to repay debt | [22–25,32,34,36,37,39] |
| | Turnover rate of total asset | Operating capacity of overall assets | [22–25,32,34,35,37,39] |
| | Growth rate of profit | Measure of the business benefits and development prospects of the enterprise | [22,25,32,35,36,39] |
| Customer service level | Rate of service satisfaction | Customer's recognition of the enterprise and satisfaction with products and services | [32–34,36,38,39,43,46] |
| | Rate of customer complaint | Customers are not satisfied with the product quality or service of the enterprise | [17,33,37,39] |
| | Speed of response | Respond quickly to unknown market demands | [17,34,40] |
| | Market share | Share of supply chain end products in the market | [32,35,38] |
| Business process | Rate of sale of marketed goods | Production and marketing operation in a certain period of time | [32,39] |
| | Rate of product qualification | Product quality level | [32,34–38] |
| | Production flexibility | Production flexibility, that is, the elasticity of the production capacity of the supply chain, can react quickly according to the market demand | [18,19,36,39] |
| | Information communication ability | Internal information sharing | [17,35,39] |
| | Logistics capability | Transportation of raw materials, storage and distribution of finished products, collaborative production and procurement, intelligent replenishment, etc. | [17,19,34,35,39] |
| Innovation and development | Rate of market forecast accuracy | Accuracy of scientific estimation of unknown market development trend | [32,34] |
| | New service development efforts | Design new or improved service concepts to meet customers' unmet needs | [17,34,35,43] |
| | Rate of R&D investment | Research and development of new products, new processes, and new materials | [27,35] |
| | Innovative ability of member learning | Stimulate and enhance the innovative consciousness of employees and improve the comprehensive quality of workers | [39,40] |
| | Proportion of scientific research personnel | Personnel specialized in scientific research and technical research | [32,34,39] |
| Green level | Environmental impact degree | Possible environmental impact of project implementation | [24,26,27,39] |
| | Environmental benefits | Proceeds from environmental protection and environmental impact reduction activities | [17,28] |
| | Rate of resource utilization | The amount of value that a certain number of resources can create | [27,28,33,37–39] |
| | Rate of waste recovery | Degree of waste recycling | [16,17,27,34,35] |
| | Green consensus | Public recognition of green concepts such as "green win-win and sustainable development" of enterprises | [34,37,39,41] |

The previous literature offers many evaluation indicators for analyzing the performance evaluation process of GSCs from different perspectives. However, due to the limited resources in the actual supply chain management process, it is unrealistic to study and

analyze all the performance evaluation indicators one by one. Thus, it is necessary to select only the critical indicators. To do so, the AHP method, gray evaluation method, and catastrophe progression method were used for quantitative analysis. These methods share a common problem, that is, the various evaluation indicators in the evaluation process are treated in isolation, without taking into account the interaction between them. For a comprehensive performance evaluation, various indicators should be understood in terms of their interaction with each other, forming a complex evaluation indicator system. Therefore, in what follows, we propose a new method of selecting critical performance evaluation indicators for GSCs.

## 3. Methodologies

### 3.1. Delphi Method

The Delphi method was initiated and implemented by the RAND Corporation in the 1950s and has the characteristics of anonymity, feedback, and statistics. It is essentially a method of anonymous inquiry by experts that involves soliciting opinions and feedback repeatedly until consensus is reached. The Delphi method adopts a back-to-back approach, which enables each expert to make independent judgments. It overcomes the subjective differences caused by different expert fields, experiences, personal cognition, etc. It is a scientific and practical analysis method [47]. When using the Delphi method to screen indicators, we first select representative scholars with professional knowledge and rich experience in decision-making issues. Anonymity ensures that the experts can freely and independently put forward their own opinions on decision-making issues. The experts are provided with as much information as possible to make judgments. Consistent opinions are sought through multiple rounds of feedback.

The specific implementation process of the Delphi method is shown in Figure 1.

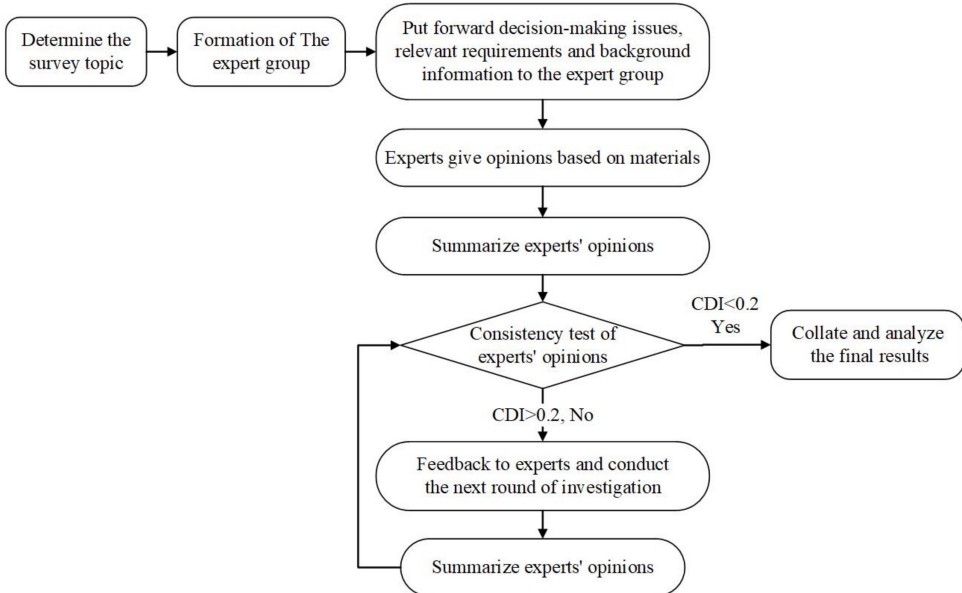

**Figure 1.** Implementation process of the Delphi method.

Implementation steps of the Delphi method:

(1) Determine the subject of investigation and draw up an outline of the investigation.
(2) Establish an expert group to determine the number of experts and the background of each expert.
(3) Present the research questions, relevant requirements, and background material to the expert group, and provide as much information as possible for the experts to make judgments.
(4) The experts make their own judgment based on this information.

(5)    Summarize the expert opinions and ensure that they are consistent.

If the experts' opinions are consistent, the final results are analyzed. If the experts' opinions are inconsistent, they are provided with feedback and additional information, and another round of investigation is conducted until the expert opinions are consistent. The consensus deviation index (CDI) was used to calculate the consensus degree of the expert panel. The CDI value was equal to the standard deviation divided by the mean. A threshold was needed in advance. If the CDI values were greater than the threshold, it indicated a significant divergence in the experts' opinions, and the next round of expert scoring is required until all the CDI values were lower than the threshold.

### 3.2. Fuzzy DEMATEL-Based ANP Model

The DEMATEL method was first proposed by A. Gabus and E. Fontela to understand complex and difficult decision-making problems in the real world. It is a systematic analysis method based on graph theory and matrix tools. When using the DEMATEL method for performance evaluation, we first analyze the composition of the indicators in the system and the logical relationship between the indicators. Then, we build a direct impact matrix. After calculations, we can obtain a comprehensive impact matrix, with which we can calculate the influence of each indicator on other indicators and determine the causality and prominence of each indicator as the basis for constructing the model. Thus, the causal relationship between indicators and the position of each indicator in the system are determined, and a causal relationship diagram of the indicators is generated [48].

The ANP method is combined with the DEMATEL method to form the DEMATEL-ANP method, which plays an important role in determining critical factors and causal relationships. The causal relationship diagram of indicators formed by the DEMATEL method lays the foundation for ANP modeling. At the same time, the total influence matrix obtained by the DEMATEL method can be directly used as an unweighted super-matrix in the ANP model, which avoids the cumbersome work of comparing two indicators in the ANP method and the problem of consistency testing. The weights of the indicators are determined by comprehensively considering the prominence calculated by the DEMATEL method and the weight calculated by ANP method. The final ranking of each indicator is thus obtained.

In our study, considering the ambiguity and uncertainty of various indicators, we explore the fuzzy DEMATEL-based ANP model for selecting key performance evaluation indicators. The specific operation steps are as follows:

First, we determined the interaction among the evaluation indicators through a questionnaire. Using the triangular blur number, we obtained the fuzzy direct influence matrix $\widetilde{Z}$, including $Z_l$, $Z_m$, $Z_u$ as the direct influence matrix of the lower limit, median, and the upper limit respectively. In Equation (1), $\widetilde{Z}_{ij}$ represents the fuzzy impact of the indicator $i$ on the indicator $j$.

$$\widetilde{Z} = \begin{bmatrix} \widetilde{z}_{11} & \widetilde{z}_{12} & \cdots & \widetilde{z}_{1n} \\ \widetilde{z}_{21} & \widetilde{z}_{22} & \cdots & \widetilde{z}_{2n} \\ \vdots & \vdots & \ddots & \vdots \\ \widetilde{z}_{n1} & \widetilde{z}_{n2} & \vdots & \widetilde{z}_{nn} \end{bmatrix} \tag{1}$$

Second, we normalized the fuzzy direct influence matrix $\widetilde{Z}$ to obtain the normalized fuzzy direct influence matrix $\widetilde{X}$. The calculation formula is shown in Equation (2), and the calculation of $\lambda$ is shown in Equation (3).

$$\widetilde{X} = \lambda \cdot \widetilde{Z} \tag{2}$$

$$\lambda = \frac{1}{\max_{1 \le i \le n} \sum_{j=1}^{n} \widetilde{z}_{ij}} (i, j = 1, 2, \cdots, n) \tag{3}$$

Third, we calculated the fuzzy total influence matrix $\widetilde{T}$ according to the normative fuzzy direct influence matrix $\widetilde{X}$. The calculation formula is shown in Equation (4), where $I$ was the unit matrix. Then, we created the crisp total influence matrix T based on $\widetilde{T}$. The crisp total influence matrix $T$ is directly used as the unweighted super-matrix of the ANP model. After the ANP operation, the weighted super-matrix and limited super-matrix were obtained successively.

$$\widetilde{T} = \widetilde{X}\left(I - \widetilde{X}\right)^{-1} \tag{4}$$

Fourth, we determined the causal relationship of the performance evaluation indicators. In the total influence matrix $T$, we used $d_i$ to represent the sum of evaluation indicators in each row and $r_i$ to represent the sum of the evaluation indicators in each column. Here, $d_i$ represents the total influence value of the $i$-th indicator on other indicators, and $r_i$ represents the total influence value of the $i$-th indicator on all other indicators. The difference between the influence degree and the affected degree of the $i$-th indicator is the causality of this indicator, which is recorded as $d_i - r_i$. If $d_i - r_i > 0$, this indicates that the impact of this indicator on other indicators is greater than that of other indicators on itself, so this indicator is called a driving factor. The greater the difference, the greater the impact of this indicator on other indicators. Conversely, if $d_i - r_i < 0$, this indicator is called a result factor. Therefore, we built a causal relationship diagram based on the classification of all indicators.

Finally, we sorted the indicators, and then determined the critical evaluation indicators. The prominence of each indicator was obtained by adding the influence degree $d_i$ and the affected degree $r_i$ of the $i$-th indicator. This represents the role of the evaluation indicator in the whole performance evaluation system, such that the importance of the indicator can be obtained by sorting the prominence. At the same time, we calculated the weight of each evaluation indicator according to the ANP model, expressed by $w_i$. Further, we calculated the weighted prominence of each evaluation indicator through the formula $w_i(d_i + r_i)$, and, finally, ranked the indicators according to the weighted prominence and determined the critical indicators in the evaluation index system.

To sum up, the key indicator selection process based on the fuzzy DEMATEL-based ANP model is shown in Figure 2.

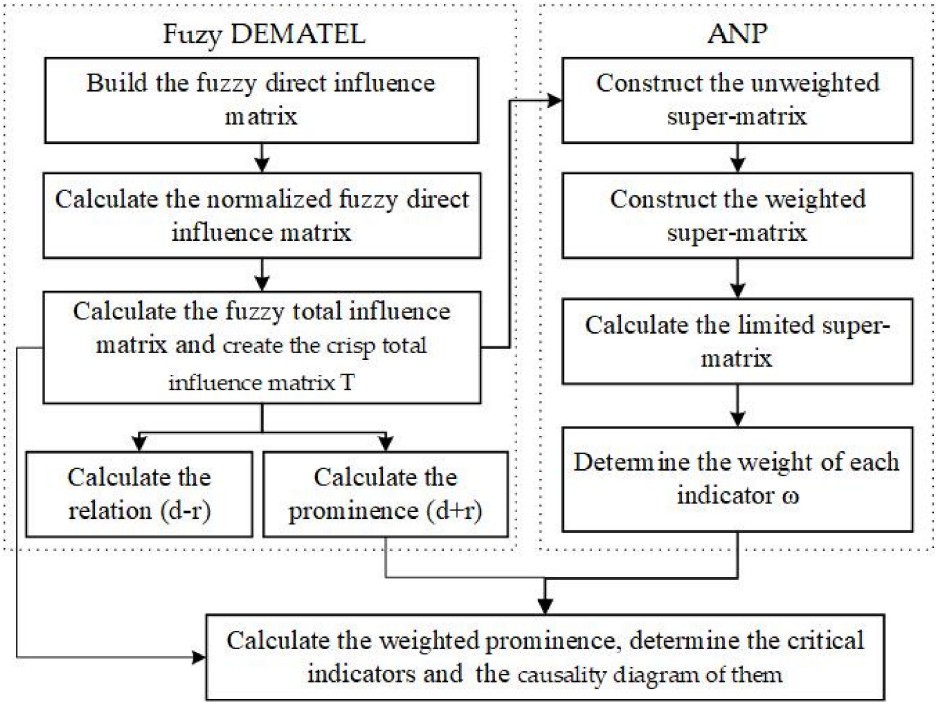

**Figure 2.** The framework of the fuzzy DEMATEL-based ANP model.

## 4. Empirical Study

### 4.1. Establishing the Formal Decision Structure Based on the Delphi Method

The Delphi method was used to screen and optimize the initial performance evaluation index system of GSCs. Six experts, with rich practical experience and theoretical background in green supply chain operation were selected, as shown in Table 2.

**Table 2.** Professional backgrounds of the selected six experts for the Delphi survey.

| Expert | Duties | Gender | Age | Specializes in | Working Area | Seniority |
|---|---|---|---|---|---|---|
| I | Professor | Male | 46 | Logistics | Shandong | 15~20 |
| II | Professor | Male | 45 | Supply chain management | Beijing | 15~20 |
| III | Associate Professor | Male | 35 | Supply chain management | Shandong | 10~15 |
| IV | Associate Research Fellow | Male | 43 | Green logistics and supply chain | Shandong | 15~20 |
| V | Senior Manager | Male | 50 | Enterprise management | Shanxi | 20~30 |
| VI | Purchasing Manager | Female | 38 | Purchasing management | Shanxi | 15~20 |

In the first round of the Delphi questionnaire, an initial research framework, as shown in Table 1, was provided to experts. Experts judged whether the listed indicators were suitable for the performance evaluation of GSCs according to their experience, and checked whether the description of the indicators was clear.

In the second round of the Delphi questionnaire, the experts scored the necessity of each indicator on a scale of 0~10. A score of 0 denoted that the indicator was absolutely unnecessary and one of 10 indicated that it was absolutely necessary. The consensus deviation index (CDI) was used to calculate the consensus degree of the expert panel. Taking 0.2 as the threshold of CDI, if it was greater than 0.2, it indicated a significant divergence in the experts' opinions, and the next round of expert scoring was required until all the CDI values were lower than 0.2. As shown in Table 3, the CDI values of 12 indicators were lower than 0.2, indicating that experts agreed on the 12 evaluation indicators. And the CDI values of the other 12 indicators were greater than 0.2. To reach a consensus, the third round of the Delphi questionnaire was conducted.

In the third round, the mean value and standard deviation of the second-round questionnaire filled out by all experts were presented. Experts who scored, in the previous round, outside the average value (plus or minus one standard deviation) were asked to provide reasons for their scores to avoid errors caused by unnecessary factors. The scoring results of the third round of the Delphi survey showed that the CDI values of 24 indicators were all less than 0.2. After the discussion, the experts agreed to take the average score of 6 as the critical value. As a result, indicators whose mean values were less than 6 were judged to be unnecessary and discarded from further consideration. A total of 12 evaluation indicators were eliminated, including the rate of return on total assets, the rate of total asset liability, the rate of total asset turnover in the financial value level, the rate of customer complaint, the speed of response in the customer service level, the rate of sale of marketed goods, and the information communication ability in the business process. The final index system for performance evaluation of GSC was shown in Table 4.

**Table 3.** Necessity analysis of indicators in the second round of Delphi questionnaire.

| Dimension | Indicator | Necessity Scoring | | | | | | Mean Value | Standard Deviation | CDI |
|---|---|---|---|---|---|---|---|---|---|---|
| | | A | B | C | D | E | F | | | |
| Financial value | Return rate of net assets | 8 | 8 | 6 | 6 | 8 | 8 | 7.3333 | 1.0328 | 0.1408 |
| | Return rate of total assets | 4 | 5 | 5 | 6 | 8 | 6 | 5.6667 | 1.3663 | 0.2411 |
| | Rate of total asset liability | 3 | 6 | 7 | 4 | 4 | 5 | 4.8333 | 1.4720 | 0.3045 |
| | Turnover rate of total asset | 5 | 8 | 4 | 6 | 6 | 4 | 5.5000 | 1.5166 | 0.2757 |
| | Growth rate of profit | 6 | 8 | 6 | 6 | 7 | 8 | 6.8333 | 0.9832 | 0.1439 |

**Table 3.** *Cont.*

| Dimension | Indicator | Necessity Scoring | | | | | | Mean Value | Standard Deviation | CDI |
|---|---|---|---|---|---|---|---|---|---|---|
| | | A | B | C | D | E | F | | | |
| Customer service level | Rate of service satisfaction | 10 | 6 | 8 | 8 | 8 | 7 | 7.8333 | 1.3292 | 0.1697 |
| | Rate of customer complaint | 5 | 8 | 5 | 6 | 8 | 7 | 6.5000 | 1.3784 | 0.2121 |
| | Speed of response | 8 | 4 | 5 | 6 | 8 | 6 | 6.1667 | 1.6021 | 0.2598 |
| | Market share | 9 | 7 | 8 | 8 | 7 | 8 | 7.8333 | 0.7528 | 0.0961 |
| Business process | Rate of sale of marketed goods | 5 | 5 | 2 | 6 | 6 | 3 | 4.5000 | 1.6432 | 0.3651 |
| | Rate of product qualification | 7 | 6 | 7 | 8 | 9 | 8 | 7.5000 | 1.0488 | 0.1398 |
| | Production flexibility | 8 | 10 | 7 | 7 | 8 | 10 | 8.3333 | 1.3663 | 0.1640 |
| | Information communication ability | 6 | 3 | 6 | 6 | 3 | 5 | 4.8333 | 1.4720 | 0.3045 |
| | Logistics capability | 8 | 9 | 8 | 10 | 9 | 9 | 8.8333 | 0.7528 | 0.0852 |
| Innovation and development | Rate of market forecast accuracy | 7 | 5 | 4 | 6 | 9 | 3 | 5.6667 | 2.1602 | 0.3812 |
| | New service development efforts | 8 | 6 | 7 | 8 | 7 | 8 | 7.3333 | 0.8165 | 0.1113 |
| | Rate of R&D investment | 7 | 9 | 6 | 6 | 7 | 8 | 7.1667 | 1.1690 | 0.1631 |
| | Innovative ability of member learning | 4 | 6 | 4 | 9 | 7 | 6 | 6.0000 | 1.8974 | 0.3162 |
| | Proportion of scientific research personnel | 7 | 6 | 9 | 6 | 3 | 7 | 6.3333 | 1.9664 | 0.3105 |
| Green level | Environmental impact degree | 6 | 5 | 2 | 5 | 7 | 4 | 4.8333 | 1.7224 | 0.3564 |
| | Environmental benefits | 6 | 9 | 7 | 8 | 5 | 6 | 6.8333 | 1.4720 | 0.2154 |
| | Rate of resource utilization | 10 | 8 | 10 | 7 | 8 | 8 | 8.5000 | 1.2247 | 0.1441 |
| | Rate of waste recovery | 8 | 7 | 7 | 7 | 10 | 9 | 8.0000 | 1.2649 | 0.1581 |
| | Green consensus | 7 | 9 | 8 | 6 | 6 | 7 | 7.1667 | 1.1690 | 0.1631 |

**Table 4.** Necessity analysis of indicators in the third round of Delphi questionnaire.

| Dimension | Indicator | Necessity Scoring | | | | | | Mean Value | Standard Deviation | CDI | Variable Number |
|---|---|---|---|---|---|---|---|---|---|---|---|
| | | A | B | C | D | E | F | | | | |
| Financial value | Return rate of net assets | 8 | 8 | 6 | 6 | 8 | 8 | 7.3333 | 1.0328 | 0.1408 | A1 |
| | Return rate of total assets | 5 | 5 | 5 | 6 | 7 | 5 | 5.5000 | 0.8367 | 0.1521 | - |
| | Rate of total asset liability | 5 | 6 | 5 | 4 | 4 | 6 | 5.0000 | 0.8944 | 0.1789 | - |
| | Turnover rate of total asset turnover | 5 | 6 | 4 | 6 | 6 | 6 | 5.5000 | 0.8367 | 0.1521 | - |
| | Growth rate of profit | 6 | 8 | 6 | 6 | 7 | 8 | 6.8333 | 0.9832 | 0.1439 | A2 |
| Customer service level | Rate of service satisfaction | 10 | 6 | 8 | 8 | 8 | 7 | 7.8333 | 1.3292 | 0.1697 | B1 |
| | Rate of customer complaint | 5 | 5 | 5 | 6 | 7 | 6 | 5.6667 | 0.8165 | 0.1441 | - |
| | Speed of response | 6 | 4 | 5 | 6 | 5 | 4 | 5.0000 | 0.8944 | 0.1789 | - |
| | Market share | 9 | 7 | 8 | 8 | 7 | 8 | 7.8333 | 0.7528 | 0.0961 | B2 |
| Business process | Rate of sale of marketed goods | 5 | 5 | 4 | 6 | 6 | 4 | 5.0000 | 0.8944 | 0.1789 | - |
| | Rate of product qualification | 7 | 6 | 7 | 8 | 9 | 8 | 7.5000 | 1.0488 | 0.1398 | C1 |
| | Production flexibility | 8 | 10 | 7 | 7 | 8 | 10 | 8.3333 | 1.3663 | 0.1640 | C2 |
| | Information communication ability | 5 | 5 | 4 | 6 | 4 | 6 | 5.0000 | 0.8944 | 0.1789 | - |
| | Logistics capability | 8 | 9 | 8 | 10 | 9 | 9 | 8.8333 | 0.7528 | 0.0852 | C3 |
| Innovation and development | Rate of market forecast accuracy | 6 | 5 | 5 | 6 | 7 | 6 | 5.8333 | 0.7528 | 0.1290 | - |
| | New service development efforts | 8 | 6 | 7 | 8 | 7 | 8 | 7.3333 | 0.8165 | 0.1113 | D1 |
| | Rate of R&D investment | 7 | 9 | 6 | 6 | 7 | 8 | 7.1667 | 1.1690 | 0.1631 | D2 |
| | Innovative ability of member learning | 5 | 6 | 4 | 6 | 6 | 7 | 5.6667 | 1.0328 | 0.1823 | - |
| | Proportion of scientific research personnel | 6 | 6 | 7 | 6 | 4 | 5 | 5.6667 | 1.0328 | 0.1823 | - |
| Green level | Environmental impact degree | 4 | 5 | 4 | 5 | 4 | 6 | 4.6667 | 0.8165 | 0.1750 | - |
| | Environmental benefits | 5 | 6 | 7 | 5 | 5 | 5 | 5.5000 | 0.8367 | 0.1521 | - |
| | Rate of resource utilization | 10 | 8 | 10 | 7 | 8 | 8 | 8.5000 | 1.2247 | 0.1441 | E1 |
| | Rate of waste recovery | 8 | 7 | 7 | 7 | 10 | 9 | 8.0000 | 1.2649 | 0.1581 | E2 |
| | Green consensus | 7 | 9 | 8 | 6 | 6 | 7 | 7.1667 | 1.1690 | 0.1631 | E3 |

### 4.2. Identification and Analysis of Critical Performance Evaluation Indicators

We used the fuzzy DEMATEL method to study the causal relationship between the indicators. The internal influence relationship of the indicators was obtained through a questionnaire. We designed 132 paired comparative questions in the questionnaire, with scores 0, 1, and 2, where 0 represents no impact, 1 represents a general impact, and 2 represents a significant impact. We applied the fuzzy linguistic scale as (0, 0, 1), (0, 1, 2), and (1, 2, 2) corresponding to 0, 1, and 2. The question posed was: "how is the impact of the return rate of net assets on the growth rate of profit". Part of the questionnaire was shown in Table A1 in Appendix A.

The questionnaire was filled out by six experts in the field of GSCs. When processing the data, the opinions of each expert were treated equally. The results of the questionnaire were summarized, and the average value was calculated. The direct influence matrix of the lower limit, the median, and the upper limit were as shown in Tables 5–7.

**Table 5.** The direct influence matrix of the lower limit.

| Indicator | A1 | A2 | B1 | B2 | C1 | C2 | C3 | D1 | D2 | E1 | E2 | E3 |
|---|---|---|---|---|---|---|---|---|---|---|---|---|
| A1 | 0.0000 | 0.8333 | 0.3333 | 0.5000 | 0.1667 | 0.5000 | 0.3333 | 0.5000 | 0.8333 | 0.3333 | 0.1667 | 0.1667 |
| A2 | 0.8333 | 0.0000 | 0.3333 | 0.8333 | 0.6667 | 0.5000 | 0.5000 | 0.5000 | 0.8333 | 0.5000 | 0.1667 | 0.3333 |
| B1 | 0.6667 | 1.0000 | 0.0000 | 1.0000 | 0.3333 | 0.3333 | 0.1667 | 0.3333 | 0.0000 | 0.1667 | 0.0000 | 0.1667 |
| B2 | 0.6667 | 0.6667 | 0.0000 | 0.0000 | 0.0000 | 0.1667 | 0.1667 | 0.3333 | 0.3333 | 0.5000 | 0.1667 | 0.3333 |
| C1 | 0.6667 | 0.8333 | 1.0000 | 0.8333 | 0.0000 | 0.0000 | 0.0000 | 0.0000 | 0.1667 | 0.3333 | 0.3333 | 0.3333 |
| C2 | 0.3333 | 0.3333 | 0.6667 | 0.1667 | 0.6667 | 0.0000 | 0.3333 | 0.3333 | 0.1667 | 0.5000 | 0.3333 | 0.0000 |
| C3 | 0.5000 | 0.5000 | 1.0000 | 0.5000 | 0.1667 | 0.5000 | 0.0000 | 0.1667 | 0.1667 | 0.3333 | 0.3333 | 0.3333 |
| D1 | 0.3333 | 0.5000 | 0.3333 | 0.5000 | 0.0000 | 0.5000 | 0.3333 | 0.0000 | 0.5000 | 0.3333 | 0.1667 | 0.3333 |
| D2 | 0.1667 | 0.5000 | 0.3333 | 0.3333 | 0.3333 | 0.3333 | 0.0000 | 0.3333 | 0.0000 | 0.3333 | 0.1667 | 0.3333 |
| E1 | 0.5000 | 0.5000 | 0.3333 | 0.1667 | 0.1667 | 0.3333 | 0.1667 | 0.3333 | 0.3333 | 0.0000 | 0.1667 | 0.5000 |
| E2 | 0.1667 | 0.3333 | 0.5000 | 0.1667 | 0.1667 | 0.5000 | 0.1667 | 0.3333 | 0.3333 | 0.6667 | 0.0000 | 0.5000 |
| E3 | 0.5000 | 0.1667 | 1.0000 | 0.8333 | 0.5000 | 0.3333 | 0.3333 | 0.6667 | 0.8333 | 0.6667 | 0.8333 | 0.0000 |

**Table 6.** The direct influence matrix of the median.

| Indicator | A1 | A2 | B1 | B2 | C1 | C2 | C3 | D1 | D2 | E1 | E2 | E3 |
|---|---|---|---|---|---|---|---|---|---|---|---|---|
| A1 | 0.0000 | 1.6667 | 1.1667 | 1.5000 | 1.0000 | 1.5000 | 1.1667 | 1.5000 | 1.8333 | 1.1667 | 0.8333 | 1.1667 |
| A2 | 1.8333 | 0.0000 | 1.1667 | 1.6667 | 1.5000 | 1.5000 | 1.3333 | 1.3333 | 1.8333 | 1.3333 | 1.0000 | 1.3333 |
| B1 | 1.5000 | 2.0000 | 0.0000 | 2.0000 | 1.1667 | 1.1667 | 0.8333 | 1.3333 | 1.0000 | 1.0000 | 0.8333 | 1.0000 |
| B2 | 1.6667 | 1.6667 | 0.8333 | 0.0000 | 0.6667 | 0.8333 | 1.0000 | 1.3333 | 1.1667 | 1.5000 | 1.0000 | 1.1667 |
| C1 | 1.6667 | 1.8333 | 2.0000 | 1.8333 | 0.0000 | 0.6667 | 0.3333 | 0.6667 | 0.6667 | 1.3333 | 1.1667 | 1.0000 |
| C2 | 1.1667 | 1.1667 | 1.6667 | 1.1667 | 1.6667 | 0.0000 | 1.3333 | 1.1667 | 1.0000 | 1.5000 | 1.3333 | 1.0000 |
| C3 | 1.5000 | 1.5000 | 2.0000 | 1.3333 | 0.6667 | 1.3333 | 0.0000 | 1.0000 | 1.0000 | 1.1667 | 1.0000 | 1.1667 |
| D1 | 1.3333 | 1.5000 | 1.3333 | 1.5000 | 0.3333 | 1.3333 | 1.0000 | 0.0000 | 1.5000 | 1.0000 | 0.8333 | 1.1667 |
| D2 | 1.0000 | 1.3333 | 1.1667 | 1.3333 | 1.1667 | 1.1667 | 0.6667 | 1.1667 | 0.0000 | 1.1667 | 0.8333 | 1.0000 |
| E1 | 1.3333 | 1.5000 | 1.1667 | 1.0000 | 1.0000 | 1.1667 | 0.8333 | 1.1667 | 1.0000 | 0.0000 | 0.8333 | 1.3333 |
| E2 | 1.0000 | 1.3333 | 1.5000 | 1.1667 | 1.0000 | 1.1667 | 0.6667 | 1.1667 | 1.3333 | 1.6667 | 0.0000 | 1.5000 |
| E3 | 1.3333 | 1.1667 | 2.0000 | 1.8333 | 1.0000 | 1.3333 | 0.8333 | 1.6667 | 1.6667 | 1.5000 | 1.8333 | 0.0000 |

**Table 7.** The direct influence matrix of the upper limit.

| Indicator | A1 | A2 | B1 | B2 | C1 | C2 | C3 | D1 | D2 | E1 | E2 | E3 |
|---|---|---|---|---|---|---|---|---|---|---|---|---|
| A1 | 0.0000 | 1.8333 | 1.8333 | 2.0000 | 1.8333 | 2.0000 | 1.8333 | 2.0000 | 2.0000 | 1.8333 | 1.6667 | 2.0000 |
| A2 | 2.0000 | 0.0000 | 1.8333 | 1.8333 | 1.8333 | 2.0000 | 1.8333 | 1.8333 | 2.0000 | 1.8333 | 1.8333 | 2.0000 |
| B1 | 1.8333 | 2.0000 | 0.0000 | 2.0000 | 1.8333 | 1.8333 | 1.6667 | 2.0000 | 2.0000 | 1.8333 | 1.8333 | 1.8333 |
| B2 | 2.0000 | 2.0000 | 1.8333 | 0.0000 | 1.6667 | 1.6667 | 2.0000 | 2.0000 | 1.6667 | 2.0000 | 1.8333 | 1.8333 |
| C1 | 2.0000 | 2.0000 | 2.0000 | 2.0000 | 0.0000 | 1.6667 | 1.3333 | 1.6667 | 1.5000 | 2.0000 | 1.8333 | 1.6667 |
| C2 | 1.8333 | 1.8333 | 2.0000 | 2.0000 | 2.0000 | 0.0000 | 2.0000 | 1.8333 | 1.8333 | 2.0000 | 2.0000 | 2.0000 |
| C3 | 2.0000 | 2.0000 | 2.0000 | 1.8333 | 1.5000 | 1.8333 | 0.0000 | 1.8333 | 1.8333 | 1.8333 | 1.6667 | 1.8333 |
| D1 | 2.0000 | 2.0000 | 2.0000 | 2.0000 | 1.3333 | 1.8333 | 1.6667 | 0.0000 | 2.0000 | 1.6667 | 1.6667 | 1.8333 |
| D2 | 1.8333 | 1.8333 | 1.8333 | 2.0000 | 1.8333 | 1.8333 | 1.6667 | 1.8333 | 0.0000 | 1.8333 | 1.6667 | 1.6667 |
| E1 | 1.8333 | 2.0000 | 1.8333 | 1.8333 | 1.8333 | 1.8333 | 1.6667 | 1.8333 | 1.6667 | 0.0000 | 1.6667 | 1.8333 |
| E2 | 1.8333 | 2.0000 | 2.0000 | 2.0000 | 1.8333 | 1.6667 | 1.5000 | 1.8333 | 2.0000 | 2.0000 | 0.0000 | 2.0000 |
| E3 | 1.8333 | 2.0000 | 2.0000 | 2.0000 | 1.5000 | 2.0000 | 1.5000 | 2.0000 | 1.8333 | 1.8333 | 2.0000 | 0.0000 |

We used Equation (2) to normalize the initial fuzzy direct influence matrix, and further calculated the fuzzy total influence matrix $\tilde{T}$ according to Equation (4). The fuzzy total influence matrix $\tilde{T}$ was shown in Tables 8–10.

**Table 8.** The total influence matrix of the lower limit.

| Indicator | A1 | A2 | B1 | B2 | C1 | C2 | C3 | D1 | D2 | E1 | E2 | E3 |
|-----------|------|------|------|------|------|------|------|------|------|------|------|------|
| A1 | 0.1381 | 0.2708 | 0.1730 | 0.2164 | 0.1105 | 0.1732 | 0.1158 | 0.1718 | 0.2358 | 0.1620 | 0.0883 | 0.1076 |
| A2 | 0.2884 | 0.2027 | 0.2106 | 0.2990 | 0.1940 | 0.1955 | 0.1525 | 0.1954 | 0.2646 | 0.2133 | 0.1077 | 0.1520 |
| B1 | 0.2312 | 0.2888 | 0.1123 | 0.2806 | 0.1252 | 0.1416 | 0.0922 | 0.1438 | 0.1210 | 0.1329 | 0.0605 | 0.1007 |
| B2 | 0.1963 | 0.2081 | 0.0962 | 0.1085 | 0.0645 | 0.1048 | 0.0786 | 0.1280 | 0.1474 | 0.1584 | 0.0750 | 0.1116 |
| C1 | 0.2410 | 0.2822 | 0.2582 | 0.2740 | 0.0832 | 0.1026 | 0.0704 | 0.1069 | 0.1485 | 0.1624 | 0.1096 | 0.1311 |
| C2 | 0.1658 | 0.1855 | 0.2036 | 0.1517 | 0.1613 | 0.0823 | 0.1016 | 0.1267 | 0.1165 | 0.1621 | 0.0988 | 0.0710 |
| C3 | 0.2109 | 0.2308 | 0.2641 | 0.2212 | 0.1085 | 0.1709 | 0.0679 | 0.1258 | 0.1387 | 0.1596 | 0.1104 | 0.1268 |
| D1 | 0.1625 | 0.1993 | 0.1539 | 0.1912 | 0.0728 | 0.1565 | 0.1052 | 0.0862 | 0.1703 | 0.1443 | 0.0801 | 0.1160 |
| D2 | 0.1252 | 0.1816 | 0.1390 | 0.1548 | 0.1086 | 0.1181 | 0.0495 | 0.1199 | 0.0854 | 0.1299 | 0.0721 | 0.1067 |
| E1 | 0.1802 | 0.1942 | 0.1525 | 0.1450 | 0.0933 | 0.1298 | 0.0801 | 0.1309 | 0.1470 | 0.0921 | 0.0794 | 0.1357 |
| E2 | 0.1405 | 0.1774 | 0.1822 | 0.1483 | 0.0970 | 0.1560 | 0.0813 | 0.1340 | 0.1456 | 0.1905 | 0.0577 | 0.1409 |
| E3 | 0.2562 | 0.2459 | 0.3107 | 0.3151 | 0.1770 | 0.1855 | 0.1348 | 0.2294 | 0.2704 | 0.2492 | 0.2036 | 0.1168 |

**Table 9.** The total influence matrix of the median.

| Indicator | A1 | A2 | B1 | B2 | C1 | C2 | C3 | D1 | D2 | E1 | E2 | E3 |
|-----------|------|------|------|------|------|------|------|------|------|------|------|------|
| A1 | 0.4784 | 0.6083 | 0.5474 | 0.5902 | 0.4152 | 0.5000 | 0.3989 | 0.5128 | 0.5410 | 0.5113 | 0.4083 | 0.4692 |
| A2 | 0.6210 | 0.5585 | 0.5886 | 0.6414 | 0.4715 | 0.5339 | 0.4345 | 0.5394 | 0.5768 | 0.5577 | 0.4477 | 0.5117 |
| B1 | 0.5481 | 0.6086 | 0.4610 | 0.5993 | 0.4108 | 0.4666 | 0.3697 | 0.4888 | 0.4813 | 0.4867 | 0.3950 | 0.4462 |
| B2 | 0.5241 | 0.5562 | 0.4805 | 0.4534 | 0.3597 | 0.4235 | 0.3658 | 0.4615 | 0.4542 | 0.4841 | 0.3805 | 0.4298 |
| C1 | 0.5375 | 0.5804 | 0.5520 | 0.5710 | 0.3299 | 0.4222 | 0.3275 | 0.4362 | 0.4451 | 0.4863 | 0.3977 | 0.4300 |
| C2 | 0.5356 | 0.5724 | 0.5655 | 0.5620 | 0.4428 | 0.4040 | 0.3980 | 0.4839 | 0.4839 | 0.5193 | 0.4272 | 0.4517 |
| C3 | 0.5422 | 0.5774 | 0.5698 | 0.5594 | 0.3814 | 0.4727 | 0.3170 | 0.4676 | 0.4765 | 0.4907 | 0.4007 | 0.4511 |
| D1 | 0.5049 | 0.5463 | 0.5055 | 0.5379 | 0.3424 | 0.4491 | 0.3568 | 0.3850 | 0.4792 | 0.4565 | 0.3711 | 0.4280 |
| D2 | 0.4605 | 0.5089 | 0.4703 | 0.5008 | 0.3670 | 0.4145 | 0.3176 | 0.4260 | 0.3669 | 0.4403 | 0.3508 | 0.3959 |
| E1 | 0.4921 | 0.5322 | 0.4852 | 0.4981 | 0.3686 | 0.4279 | 0.3368 | 0.4393 | 0.4397 | 0.3857 | 0.3618 | 0.4257 |
| E2 | 0.5066 | 0.5590 | 0.5357 | 0.5415 | 0.3933 | 0.4558 | 0.3495 | 0.4689 | 0.4866 | 0.5109 | 0.3375 | 0.4628 |
| E3 | 0.6020 | 0.6354 | 0.6394 | 0.6591 | 0.4503 | 0.5322 | 0.4129 | 0.5656 | 0.5756 | 0.5747 | 0.4981 | 0.4436 |

**Table 10.** The total influence matrix of the upper limit.

| Indicator | A1 | A2 | B1 | B2 | C1 | C2 | C3 | D1 | D2 | E1 | E2 | E3 |
|-----------|------|------|------|------|------|------|------|------|------|------|------|------|
| A1 | 1.9823 | 2.0603 | 2.0306 | 2.0677 | 1.8490 | 1.9565 | 1.8235 | 1.9997 | 1.9709 | 1.9893 | 1.8987 | 1.9843 |
| A2 | 2.0257 | 2.0249 | 2.0312 | 2.0615 | 1.8498 | 1.9571 | 1.8239 | 1.9934 | 1.9716 | 1.9900 | 1.9062 | 1.9850 |
| B1 | 2.0040 | 2.0526 | 1.9801 | 2.0530 | 1.8361 | 1.9357 | 1.8035 | 1.9855 | 1.9571 | 1.9752 | 1.8920 | 1.9634 |
| B2 | 1.9968 | 2.0382 | 2.0019 | 1.9956 | 1.8160 | 1.9153 | 1.8048 | 1.9716 | 1.9296 | 1.9681 | 1.8786 | 1.9497 |
| C1 | 1.9264 | 1.9664 | 1.9379 | 1.9667 | 1.7234 | 1.8473 | 1.7137 | 1.8884 | 1.8542 | 1.8991 | 1.8126 | 1.8740 |
| C2 | 2.0607 | 2.1038 | 2.0802 | 2.1110 | 1.8948 | 1.9543 | 1.8683 | 2.0346 | 2.0051 | 2.0381 | 1.9525 | 2.0258 |
| C3 | 1.9695 | 2.0104 | 1.9814 | 2.0039 | 1.7845 | 1.8961 | 1.7377 | 1.9380 | 1.9103 | 1.9344 | 1.8462 | 1.9231 |
| D1 | 1.9559 | 1.9964 | 1.9676 | 1.9969 | 1.7651 | 1.8829 | 1.7546 | 1.8887 | 1.9041 | 1.9140 | 1.8334 | 1.9098 |
| D2 | 1.9327 | 1.9728 | 1.9443 | 1.9802 | 1.7711 | 1.8668 | 1.7395 | 1.9082 | 1.8447 | 1.9050 | 1.8179 | 1.8866 |
| E1 | 1.9333 | 1.9804 | 1.9450 | 1.9739 | 1.7716 | 1.8676 | 1.7400 | 1.9088 | 1.8745 | 1.8696 | 1.8186 | 1.8943 |
| E2 | 2.0035 | 2.0523 | 2.0226 | 2.0527 | 1.8359 | 1.9285 | 1.7961 | 1.9784 | 1.9567 | 1.9818 | 1.8558 | 1.9700 |
| E3 | 1.9918 | 2.0403 | 2.0109 | 2.0408 | 1.8113 | 1.9311 | 1.7860 | 1.9738 | 1.9388 | 1.9634 | 1.8878 | 1.9159 |

On the basis of the fuzzy total influence matrix $\tilde{T}$, we created the crisp total influence matrix T as shown in Table 11.

We summarized and summed the row and column elements in the crisp total influence matrix to obtain the influence degree d and the affected degree r of each indicator. We then calculated the prominence degree (d + r) and the cause degree (d − r) of each indicator, and finally determined the type of each indicator, as shown in Table 12.

**Table 11.** The crisp total influence matrix of indicators.

| Indicator | A1 | A2 | B1 | B2 | C1 | C2 | C3 | D1 | D2 | E1 | E2 | E3 |
|-----------|------|------|------|------|------|------|------|------|------|------|------|------|
| A1 | 0.8663 | 0.9798 | 0.9170 | 0.9581 | 0.7916 | 0.8766 | 0.7794 | 0.8948 | 0.9159 | 0.8876 | 0.7984 | 0.8537 |
| A2 | 0.9784 | 0.9287 | 0.9435 | 1.0006 | 0.8384 | 0.8955 | 0.8036 | 0.9094 | 0.9377 | 0.9203 | 0.8205 | 0.8829 |
| B1 | 0.9277 | 0.9833 | 0.8511 | 0.9777 | 0.7907 | 0.8479 | 0.7551 | 0.8727 | 0.8531 | 0.8649 | 0.7825 | 0.8367 |
| B2 | 0.9057 | 0.9342 | 0.8596 | 0.8525 | 0.7468 | 0.8145 | 0.7497 | 0.8537 | 0.8438 | 0.8702 | 0.7780 | 0.8304 |
| C1 | 0.9017 | 0.9430 | 0.9160 | 0.9372 | 0.7122 | 0.7907 | 0.7039 | 0.8105 | 0.8159 | 0.8493 | 0.7733 | 0.8117 |
| C2 | 0.9207 | 0.9539 | 0.9498 | 0.9416 | 0.8329 | 0.8136 | 0.7893 | 0.8817 | 0.8685 | 0.9065 | 0.8262 | 0.8495 |
| C3 | 0.9075 | 0.9395 | 0.9385 | 0.9282 | 0.7582 | 0.8466 | 0.7075 | 0.8438 | 0.8419 | 0.8616 | 0.7858 | 0.8337 |
| D1 | 0.8744 | 0.9140 | 0.8757 | 0.9087 | 0.7268 | 0.8295 | 0.7389 | 0.7866 | 0.8512 | 0.8383 | 0.7615 | 0.8179 |
| D2 | 0.8395 | 0.8878 | 0.8512 | 0.8786 | 0.7489 | 0.7998 | 0.7022 | 0.8180 | 0.7657 | 0.8251 | 0.7469 | 0.7964 |
| E1 | 0.8685 | 0.9023 | 0.8609 | 0.8723 | 0.7445 | 0.8084 | 0.7190 | 0.8264 | 0.8204 | 0.7825 | 0.7533 | 0.8186 |
| E2 | 0.8835 | 0.9296 | 0.9135 | 0.9142 | 0.7754 | 0.8468 | 0.7423 | 0.8604 | 0.8630 | 0.8944 | 0.7503 | 0.8579 |
| E3 | 0.9500 | 0.9739 | 0.9870 | 1.0050 | 0.8129 | 0.8829 | 0.7779 | 0.9230 | 0.9283 | 0.9291 | 0.8632 | 0.8254 |

**Table 12.** Analysis of prominence and causality of indicators.

| Indicator | d | r | d + r | d − r | Type |
|-----------|---------|---------|---------|---------|------|
| A1 | 10.5191 | 10.8239 | 21.3430 | −0.3048 | Result factor |
| A2 | 10.8594 | 11.2699 | 22.1293 | −0.4104 | Result factor |
| B1 | 10.3437 | 10.8637 | 21.2074 | −0.5201 | Result factor |
| B2 | 10.0390 | 11.1746 | 21.2137 | −1.1356 | Result factor |
| C1 | 9.9653 | 9.2792 | 19.2445 | 0.6861 | Driving factor |
| C2 | 10.5342 | 10.0529 | 20.5870 | 0.4813 | Driving factor |
| C3 | 10.1926 | 8.9688 | 19.1614 | 1.2238 | Driving factor |
| D1 | 9.9235 | 10.2809 | 20.2044 | −0.3574 | Result factor |
| D2 | 9.6601 | 10.3053 | 19.9654 | −0.6452 | Result factor |
| E1 | 9.7770 | 10.4296 | 20.2066 | −0.6527 | Result factor |
| E2 | 10.2313 | 9.4399 | 19.6712 | 0.7913 | Driving factor |
| E3 | 10.8584 | 10.0148 | 20.8732 | 0.8436 | Driving factor |

According to the analysis of the results in Table 12, we divided the indicators into two categories: driving factors and result factors. The driving factors included the rate of product qualification (C1), production flexibility (C2), logistics capability (C3), the rate of waste recovery (E2), and green consensus (E3). These five indicators constitute the direct elements of the performance evaluation of GSCs.

We took the crisp total influence matrix in Table 11 as the unweighted super-matrix of the ANP model. The structure of the ANP model was shown in Figure 3.

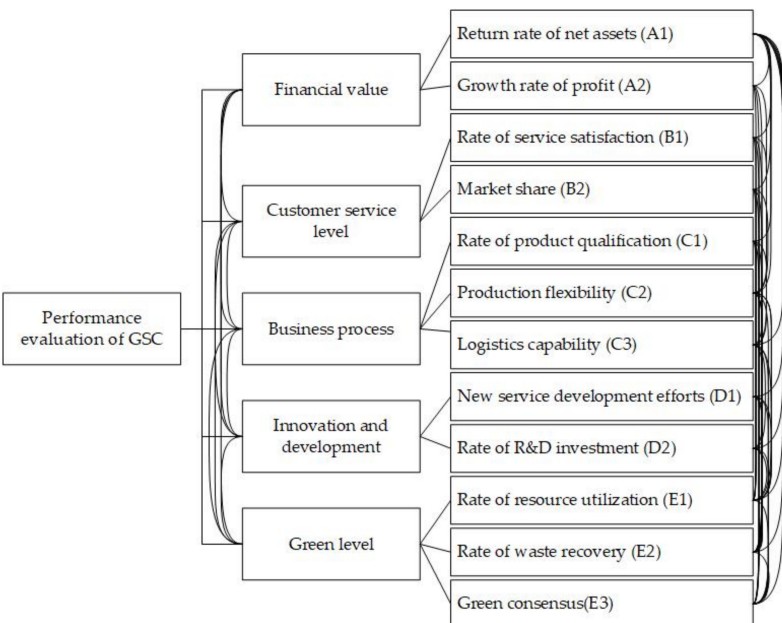

**Figure 3.** The structure of the ANP model.

Through further calculation, we derived a weighted super-matrix and a limited super-matrix. The weight coefficient of each indicator is shown in Table 13.

**Table 13.** The limited super-matrix of performance evaluation indicators of GSCs.

| Indicator | A1 | A2 | B1 | B2 | C1 | C2 | C3 | D1 | D2 | E1 | E2 | E3 |
|---|---|---|---|---|---|---|---|---|---|---|---|---|
| A1 | 0.0856 | 0.0856 | 0.0856 | 0.0856 | 0.0856 | 0.0856 | 0.0856 | 0.0856 | 0.0856 | 0.0856 | 0.0856 | 0.0856 |
| A2 | 0.0884 | 0.0884 | 0.0884 | 0.0884 | 0.0884 | 0.0884 | 0.0884 | 0.0884 | 0.0884 | 0.0884 | 0.0884 | 0.0884 |
| B1 | 0.0841 | 0.0841 | 0.0841 | 0.0841 | 0.0841 | 0.0841 | 0.0841 | 0.0841 | 0.0841 | 0.0841 | 0.0841 | 0.0841 |
| B2 | 0.0817 | 0.0817 | 0.0817 | 0.0817 | 0.0817 | 0.0817 | 0.0817 | 0.0817 | 0.0817 | 0.0817 | 0.0817 | 0.0817 |
| C1 | 0.0810 | 0.0810 | 0.0810 | 0.0810 | 0.0810 | 0.0810 | 0.0810 | 0.0810 | 0.0810 | 0.0810 | 0.0810 | 0.0810 |
| C2 | 0.0858 | 0.0858 | 0.0858 | 0.0858 | 0.0858 | 0.0858 | 0.0858 | 0.0858 | 0.0858 | 0.0858 | 0.0858 | 0.0858 |
| C3 | 0.0829 | 0.0829 | 0.0829 | 0.0829 | 0.0829 | 0.0829 | 0.0829 | 0.0829 | 0.0829 | 0.0829 | 0.0829 | 0.0829 |
| D1 | 0.0808 | 0.0808 | 0.0808 | 0.0808 | 0.0808 | 0.0808 | 0.0808 | 0.0808 | 0.0808 | 0.0808 | 0.0808 | 0.0808 |
| D2 | 0.0786 | 0.0786 | 0.0786 | 0.0786 | 0.0786 | 0.0786 | 0.0786 | 0.0786 | 0.0786 | 0.0786 | 0.0786 | 0.0786 |
| E1 | 0.0796 | 0.0796 | 0.0796 | 0.0796 | 0.0796 | 0.0796 | 0.0796 | 0.0796 | 0.0796 | 0.0796 | 0.0796 | 0.0796 |
| E2 | 0.0832 | 0.0832 | 0.0832 | 0.0832 | 0.0832 | 0.0832 | 0.0832 | 0.0832 | 0.0832 | 0.0832 | 0.0832 | 0.0832 |
| E3 | 0.0883 | 0.0883 | 0.0883 | 0.0883 | 0.0883 | 0.0883 | 0.0883 | 0.0883 | 0.0883 | 0.0883 | 0.0883 | 0.0883 |

We combined the prominence of each indicator in Table 12 with the weight coefficient in Table 13 to obtain the weighted prominence of each indicator. The results were sorted to obtain a comprehensive ranking of the performance evaluation indicators of GSCs, as shown in Table 14.

**Table 14.** Comprehensive ranking of performance evaluation indicators of GSCs.

| Indicator | Prominence | Weight | Weighted Prominence | Sorting |
|---|---|---|---|---|
| Return rate of net assets (A1) | 12.3234 | 0.0856 | 1.827 | 3 |
| Growth rate of profit (A2) | 13.3160 | 0.0884 | 1.956 | 1 |
| Rate of service satisfaction (B1) | 12.1509 | 0.0841 | 1.785 | 4 |
| Market share (B2) | 12.0725 | 0.0817 | 1.734 | 6 |
| Rate of product qualification (C1) | 10.2435 | 0.0810 | 1.559 | 12 |
| Production flexibility (C2) | 11.3400 | 0.0858 | 1.765 | 5 |
| Logistics capability (C3) | 10.0083 | 0.0829 | 1.588 | 10 |
| New service development efforts (D1) | 11.0306 | 0.0808 | 1.632 | 8 |
| Rate of R&D investment (D2) | 10.8937 | 0.0786 | 1.570 | 11 |
| Rate of resource utilization (E1) | 11.0897 | 0.0796 | 1.609 | 9 |
| Rate of waste recovery (E2) | 10.3767 | 0.0832 | 1.637 | 7 |
| Green consensus (E3) | 11.9254 | 0.0883 | 1.843 | 2 |

According to the final ranking results in Table 14, the six most critical indicators in the performance evaluation of GSC were as follows: the rate of profit growth (A2), the green consensus of GSCs (E3), the return rate of net assets (A1), the rate of service satisfaction (B1), the production flexibility (C2), and market share (B2).

Combining the crisp total influence matrix in Table 11 and the weighted prominence in Table 14, a causal relationship diagram of the six critical indicators was generated, as shown in Figure 4. In Figure 4, we took the relation (d − r) as the vertical axis and the weighted prominence (d + r) as the horizontal axis. We determined the most influential indicator for each critical indicator from the crisp total influence matrix. For example, (E3→B1) indicated that E3 affected B1 the most. As shown in Figure 4, E3 affected B1 and B2 the most. A2 affected A1, C2 and E3 the most. B1 affected A2 the most. Since E3 was the critical driving factor, the improvement of E3 would improve B1, then A2 would be improved by B1, and E3 would be improved by A2, forming a virtuous circle as 'E3→B1→A2→E3'. The performance of the overall GSC could be improved.

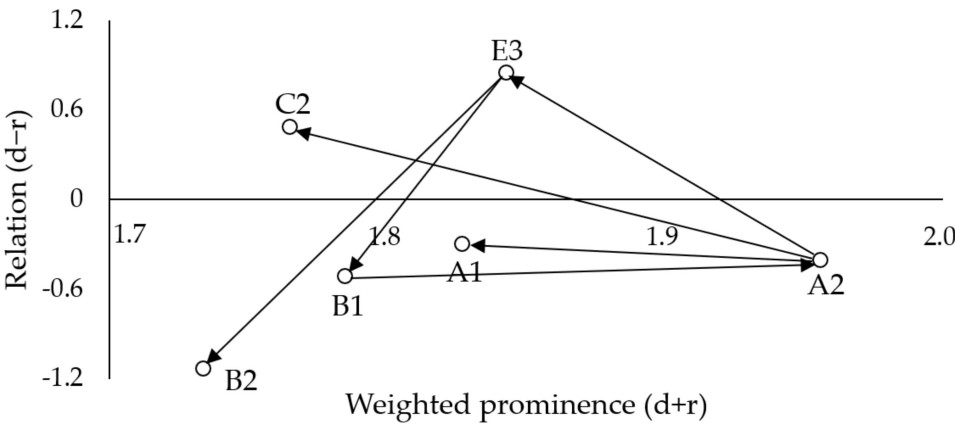

**Figure 4.** Causality diagram of critical indicators.

*4.3. Analysis of the Implementation Effect of the Case Enterprise*

Since China put forward the "Double Carbon" target in 2020, coal mining enterprises, which were traditionally high energy-consuming and high-emissions, had been under pressure to conform with green transformation. Achieving green design and green procurement of coal from mines, green mining and green transportation of coal, and ultimately achieving the goal of green mine construction were the only way for coal enterprises to achieve green transformation.

Company A was an enterprise with coal mining as its main business under Jinneng Holding Group, which paid special attention to improving the performance of the GSC in the practice of the construction of eco-friendly mines. It had promoted the construction of a GSC from various aspects such as green procurement, green mining, and green transportation of coal. In the previous process of GSC construction, due to the inadequate understanding of the key indicators affecting the performance of the GSC, the managerial measures were not targeted, and the implementation was not satisfactory. Referring to the conclusions of our study, Company A strengthened the green consensus in the process of selecting raw material suppliers and coal transportation companies. Due to the improvement of green consensus, the rate of service satisfaction and market share were significantly improved. The improvement of the service satisfaction level made the growth rate of profit increase by 5%. Meanwhile, the growth of profit further enhanced the green consensus, forming a positive cycle. In addition, the profit enhancement also promoted the optimization of the return rate on net assets and production flexibility. All in all, the construction of a GSC in Company A was more targeted.

## 5. Discussion and Implications

The empirical results showed that financial value, customer service level, business processes, and the green level were important dimensions of GSC performance evaluations. Critical performance evaluation indicators included the return rate of net assets (A1), the growth rate of profit (A2), the rate of service satisfaction (B1), market share (B2), production flexibility (C2), and the green consensus (E3). We divided the indicators in the evaluation index system into driving factors and result factors by calculating the cause degree. The detailed analysis of the results are as follows:

(1)  The production flexibility (C2) and the green consensus (E3) are the critical driving factors. The latter (E3) has a decisive impact on the rate of resource utilization (E1), the rate of waste recovery (E2), the rate of service satisfaction (B1), market share (B2), and new service development efforts (D1).

(2)  The rate of profit growth (A2) has an important impact on the financial value performance of GSC performance evaluations. The profit growth rate is an important indicator that reflects the operating efficiency and development prospects of enterprises. It further reflects the development potential of enterprises. However, the

profit growth rate is a result factor that is mainly affected by the green degree of the enterprise. The public's high recognition of GSC helps to improve the image, popularity, and influence of the green demonstration enterprise, thus bringing economic benefits. Enterprises should adhere to the principles of green development over a sustained period.

(3)    The return rate of net assets (A1) is a critical factor in performance evaluations of GSC enterprises. It reflects the input–output level and profit quality of enterprises in a certain production cycle, and it further reflects the level of investment income of enterprises. It is the most comprehensive financial indicator when studying the operation status of enterprises and measuring the asset structure and operating capacity of enterprises, and it has an important impact on performance evaluations of supply chains. However, the return on net assets is a result factor that is mainly affected by the profit growth rate.

(4)    The rate of service satisfaction (B1) plays an important role in performance evaluations of GSCs. Customers are located at the end of the supply chain and provide objective feedback for the operation effect of the enterprise supply chain. Service satisfaction reflects customers' recognition of the enterprise and satisfaction with products and services, which is the key factor when measuring customer performance evaluation. It can be seen from Figure 1 that service satisfaction is a result factor that is mainly affected by the green consensus of GSCs.

(5)    Market share (B2) is a critical indicator in GSC performance evaluation, reflecting the share of supply chain end products in the market and the product service capability of enterprises. However, the market share is a result factor that is mainly affected by the green consensus of GSCs. That is, those enterprises with a high green consensus of GSCs also have a large share of their products at the end of the supply chain.

Based on the above analysis, we propose the following managerial implications to enhance the performance of GSCs.

To improve the performance of GSCs, enterprises should basically focus on the green consensus by strengthening the selection and evaluation of supply chain partners. Partners who accept and implement the green concept should be selected and encouraged. With the improvement of the green consensus, the service satisfaction level and the market share will all be optimized directly. The improvement of the service satisfaction level can lead to an improvement of the growth rate of profit. With the improvement of the growth rate of profit, participants of the GSC can utilize more financial resources to strengthen the green consensus. These three indicators form a virtuous circle. Additionally, the production flexibility and the return rate of net assets will be directly improved by the increasing financial support. That is, the whole GSC system will be improved through the improvement of green consensus and the virtuous circle.

## 6. Conclusions

Based on the improved BSC-SCOR model, we constructed a GSC performance evaluation index system to ensure comprehensiveness and performance of the evaluation process. Then, according to the fuzzy DEMATEL-based ANP model, we selected critical indicators for evaluating GSC performance, and we proposed corresponding countermeasures according to the empirical research results. Our conclusions are as follows:

First, we sorted out the existing research results in this field, and on that basis, we reviewed an initial index system for performance evaluation of GSCs. The initial indicator set comprised 24 evaluation indicators in five dimensions: financial value, customer service level, business processes, innovation and development, and the green level. The initial indicators were screened and optimized based on an expert questionnaire survey. After calculating the average score and CDI values of each indicator, 12 unnecessary indicators were eliminated, and a GSC performance evaluation indicator system with 12 indicators in five dimensions was ultimately obtained.

Second, we used the fuzzy DEMATEL method to calculate the prominence and causality of the indicators and analyzed the causal relationship between the indicators. Six experts with rich practical experience and professional theoretical backgrounds in the field of GSCs were invited to complete the questionnaire. By summarizing the data and calculating average values, an initial fuzzy direct influence matrix was obtained. The fuzzy and crisp total influence matrices were further calculated. The indicators were divided into driving factors and result factors according to the relationship. The driving factors included the product qualification rate, production flexibility, the logistics capacity, the rate of waste recovery, and the green consensus.

Third, we selected critical indicators for evaluating GSC performance according to the fuzzy DEMATEL-based ANP model. The weighted prominence of each indicator was obtained by combining the prominence with the weight coefficient calculated by the ANP model. Then, we ranked the calculation results to determine the critical evaluation indicators in GSC performance evaluation.

Finally, according to the analysis results, some countermeasures and suggestions were put forward regarding the process of evaluating GSC performance.

Compared with previous studies [5,11,31,49], four basic dimensions of the index system were the same, including financial value, customer service, business process, and innovation and development. We all took the growth rate of profit as a typical reflection of financial performance, the rate of service satisfaction as an important reflection of customer service level, and we all emphasized the importance of logistics capability in the supply chain. In the dimension of innovation and development, we all considered new service development and R&D investment. The differences lie in the following two aspects: firstly, our study fully considered the characteristics of a GSC, adding the green level as the fifth dimension. The green level dimension included three indicators: the rate of resource utilization, the rate of waste recovery, and the green consensus. Secondly, considering the complexity of the business process of a GSC, we included the SCOR model to optimize and adjust the business process dimension. We integrated the three indicators of time flexibility, variety flexibility, and quantity flexibility into production flexibility. At the same time, the logistics capability was incorporated into the business process dimension.

In terms of the identification of critical indicators, Shafiee et al. [5] and Wang et al. [31] obtained the critical indicators from three dimensions: financial, customer service, and business process, using the gray clustering and fuzzy comprehensive evaluation methods. Jia et al. [49] identified that customer service and logistics capacity were the two most important indicators, using the extension goodness evaluation method. These conclusions are consistent with our study. Unlike from the previous studies, we not only extracted critical indicators, but also identified the critical driving indicators through the interaction analysis of any two indicators. We considered the green consensus as the most important indicator to enhance the overall performance of a GSC. We also discovered the virtuous circle among the green consensus, the service satisfaction level, and the growth rate of profit. Only if all the participants in the whole supply chain wholly accept the green development concept, can they maintain the synergy and consistency in their business and drive the improvement of financial indicators and customer satisfaction.

There were some limitations to this study. Although we classified the 24 indicators into five dimensions according to the BSC-SCOR joint model, we did not consider an exhaustive set of indicators. Such indicators should be checked one by one in consideration of innovations in supply chain models and recent research, and the selection of evaluation indicators should be further expanded. In addition, when processing the data, we treated the opinions of each expert equally, even though the theoretical experience of each expert differed. In future research, we will consider a weighted ratio to distinguish the experience of the experts.

**Author Contributions:** Conceptualization, C.Z. and J.Z.; methodology, C.Z.; software, L.T.; validation, C.Z., J.Z. and L.T.; formal analysis, C.Z.; investigation, L.T.; resources, J.Z.; data curation, L.T.; writing—original draft preparation, L.T.; writing—review and editing, C.Z.; visualization, C.Z.; supervision, J.Z.; project administration, J.Z.; funding acquisition, J.Z. All authors have read and agreed to the published version of the manuscript.

**Funding:** This research was funded by The Project of Cultivation for young top-motch Talents of Beijing Municipal Institutions under Grant BPHR202203236, and the operating funding for Beijing Key Lab of Big Data Decision Making for Green Development under grant 5026023502.

**Institutional Review Board Statement:** The study was conducted according to the guidelines of the Declaration of Helsinki, and approved by the Institutional Review Board of the School of Economics and Management, Beijing Information Science and Technology University (31 March 2023).

**Informed Consent Statement:** Informed consent was obtained from all subjects involved in the study.

**Data Availability Statement:** All data that support the findings of this study are available from the corresponding author upon reasonable request.

**Conflicts of Interest:** The authors declare no conflict of interest.

## Appendix A

**Table A1.** Part of the questionnaire.

---

1. What do you think about the impact of the return rate of net assets on its growth rate of profit?

- no impact (0)
- a general impact (1)
- a significant impact (2)

---

2. What do you think about the impact of the return rate of net assets on its rate of service satisfaction?

- no impact (0)
- a general impact (1)
- a significant impact (2)

---

3. What do you think about the impact of the return rate of net assets on its market share?

- no impact (0)
- a general impact (1)
- a significant impact (2)

---

4. What do you think about the impact of the return rate of net assets on its rate of product qualification?

- no impact (0)
- a general impact (1)
- a significant impact (2)

---

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
