# Peer review of "Identifying Critical Indicators in Performance Evaluation of Green Supply Chains Using Hybrid Multiple-Criteria Decision-Making"

_sustainability, doi:10.3390/su15076095_

Round 1

Reviewer 1 Report

The study used Delphi method and DEMATEL-ANP multi-criteria decision-making model, for developing a framework evaluating GSC performance. Few areas of improvement are:

- Choice of GSC performance indicators in Table 1 needs justification.

-Details for Delphi processes particularly 'consistency test of experts" to be added. Fig. 1 offers a high-level process flow only.

-Theoretical and practical implications of the findings need to be presented clearly.

Reviewer 2 Report

My Comments and Suggestions for the Authors are given below:

1.     The authors have prepared a study on the Performance evaluation of green supply chain (GSC). In the study, using the Delphi method at the superficial level, DEMATEL - ANP multi-criteria decision-making methods were used.

2.     As can be seen from the literature research conducted by the authors, there are many models in Performance Evaluation problems where multi-criteria decision-making methods are used either individually or in combination with several methods. Therefore the paper does not propose any new framework. The research gap that the study is trying to fill should be well-defined.

3.     In practice, the Delphi survey does not seem realistic, survey questions, etc. should be developed comprehensively. In addition, detailed information should be given about the areas of expertise of the experts, who they are, and their backgrounds. These Expert opinions' contribution to the study was limited and inadequate.

4.     The authors should provide a detailed explanation of the paired comparative questions in the questionnaire, which is stated to be designed in Section 4.2.

5.     The relationships in Figure 3 need to be explained in detail.

6.     A Structure of the ANP model should be created graphically.

7.     Critical Indicators determined by the recommended combined BSC-SCOR should be verified by testing in real logistics companies. How practitioners can benefit from this model should be demonstrated with at least one application.

Reviewer 3 Report

The authors proposed an integrated Multi-Criteria Decision-Making model to capture and evaluate the critical indicators in performace on green supply chain. The proposed model utilizes Analytical Network Process and Decision-Making Trial and Evaluation Laboratory in this regard. It is interesting topic. However, there are some issues that should be revised by the authors.

- The novelty of this paper should be further justified and to establish the contributions to the new body of knowledge.
- Abstract section should be improved.
- It can be noticed that the way of using of multi-criteria decision-making in this manuscript have been studied and applied for years in the previous literatures. So I suggest to the authors to use a recent method of MCDM like fuzzy DEMATEL, intuitionistic Fuzzy ANP or IF DEMATEL to capture the ambiguity and uncertain of various indicators used in this study…
- The presentation of the results is not enough; it should be highlighted.
- It is very interesting to compare the obtained results with other studies or methodologies.
- Managerial insights should be added to the paper. In other words, the authors should be presented no-included mathematical phrases that can help the readers who use the results, directly, as a managerial tool.

Round 2

Reviewer 2 Report

Since the subject discussed in the article is being studied extensively by many researchers, the study can partially contribute to the advancement of existing knowledge in terms of novelty aspect and the methods used in the study. However, the authors have given appropriate answers to the questions by making the necessary corrections one by one on each of the topics I have mentioned in my previous review.

I suggest to accept this paper in the present form.

Reviewer 3 Report

Thank you, I suggest to accept this paper in the current form